# Synergistic Hypolipidemic and Immunomodulatory Activity of Lactobacillus and *Spirulina platensis*

**Ragaa A. Hamouda** [1,2], **Hanafy A. Hamza** [2], **Mohammed L. Salem** [3], **Shymaa Kamal** [4], **Reem Hasaballah Alhasani** [5], **Ifat Alsharif** [6], **Hoda Mahrous** [2] and **Asmaa Abdella** [2,*]

[1] Biology Department, College of Science and Arts at Khulis, University of Jeddah, Jeddah 21921, Saudi Arabia; ragaa.hamouda@gebri.usc.edu.eg

[2] Genetic Engineering and Biotechnology Research Institute (GEBRI), University of Sadat City, Sadat City 32897, Egypt; hanafi.hamza@gebri.usc.edu.eg (H.A.H.); hoda.mahrous@gebri.usc.edu.eg (H.M.)

[3] Immunology and Biotechnology Unit, Zoology Department, Faculty of Science, Tanta University, Tanta 31111, Egypt; mohamed.labib@science.tanta.edu.eg

[4] National Organization for Drug Control and Research, Cairo 12561, Egypt; shymaa5kamal@yahoo.com

[5] Department of Biology, Faculty of Applied Science, Umm Al-Qura University, Makkah 21961, Saudi Arabia; rhhasani@uqu.edu.sa

[6] Department of Biology, Jamoum University College, Umm Al-Qura University, Makkah 21955, Saudi Arabia; eesharif@uqu.edu.sa

* Correspondence: asmaa.abdelaal@gebri.usc.edu.eg

**Abstract:** Hyperlipidemia is a significant risk factor for atherosclerosis and coronary heart disease (CHD). The aim of this study was to investigate hypolipidemic effects of Lactobacillus, Spirulina and their combination on Swiss albino mice fed a regular or high-cholesterol diet. Rosuvastatin was used as a reference drug The highest body weight, total cholesterol (TC), triglycerides (TG), low-density lipoprotein cholesterol LDL-C and the lowest high-density lipoprotein cholesterol were recorded in a positive control group (G5). Treatment with Lactobacillus or Spirulina or by their combination resulted in a significant decrease in body weight, TC, TG, LDL-C and significant increase in HDL-C ($p < 0.05$) in both mice fed a regular diet or high-cholesterol diet. The treatments induced a significant increase in Hb, MCHC and HCT levels in mice fed a regular diet ($p < 0.05$). They did not induce a significant effect on these parameters in mice fed a high-cholesterol diet, while treatment with standard rosuvastatin induced a significant decrease in these parameters ($p < 0.05$). The treatments induced a significant increase in the platelet count and WBC number in mice fed a regular diet $p < 0.05$), while they induced significant decrease in these parameters in mice fed a high-cholesterol diet $p < 0.05$. They also stimulated the innate immunity represented by both monocyte and neutrophil cells in mice fed a regular diet, while this immunity was reduced in mice fed a high-cholesterol diet. It also caused a marked reduction in inflammation and an improvement in the congestion of cardiac tissues, the aorta, and the spleen. The treatment of hyperlipidemic mice with combination of Lactobacillus and Spirulina gave similar results to those obtained with treatment by rosuvastatin.

**Keywords:** hypolipidemic; immunomodulatory; *Lactobacillus casei*; *Lactobacillus plantarum*; *Spirulina platensis*

## 1. Introduction

Cardiovascular disease (CVD) and its consequences are the leading causes of death worldwide. According to a World Health Organization estimate, cardiovascular disease is responsible for 31% of all fatalities worldwide [1]. Hyperlipidemia is a significant risk factor for atherosclerosis and coronary heart disease (CHD). Even though there are numerous risk factors for CHD, hyperlipidemia remains a primary determinant of this disease [2]. To decrease the incidence of CHD, it is necessary to reduce the level of serum cholesterol in hypercholesterolemic subjects. The most well-known hypercholesteremic medications are

statins. However, statin side effects such as myalgia, physical weakness, weariness, low energy and hyperglycemia have been reported [3].

Probiotics are defined as selected, live, microbial dietary supplements that, when supplied in appropriate amounts, have a favorable effect on the human organism through their activities in the intestinal tract [4]. Many studies demonstrated that probiotics or products containing them have a variety of health advantages, including the prevention of CVD and improving overall health [5]. According to recent studies, functional meals created with probiotic yeasts reduce the levels of lipids in the serum of rats fed a high-cholesterol diet [6]. In fact, compared to placebo-treated animals, mice and rats fed probiotic strains, such as Lactobacillus and Bifidobacterium, demonstrated decreased weight growth, fat formation, and white adipose tissue in over 85 percent of investigations [7]. In addition to improving the digestive system, probiotics are also found to have immunomodulatory effects, promoting endogenous host defense mechanisms, and resulting in gut microbiota stabilization [8]. Lactobacillus bacteria have been identified to strengthen the intestine's immunologic barrier by enhancing humoral immune responses. Probiotics and prebiotics may have immunomodulatory properties by changing the gut microbial population and reducing the activity of pathogenic intestinal microorganisms such as *Klebsiella pneumoniae* and *Clostridia perfringens* [9].

Blue–green algae (Cyanobacteria) are a group of Gram-negative photoautotrophic prokaryotes containing a blue–green-colored pigment (c-phycocyanin) [10]. Recent studies have used cyanobacterial biomasses to boost the functional product qualities of fermented milk and promote probiotic viability and nutritional features by adding a microalgae Spirulina to it [11]. Spirulina is a blue–green microalga that is rich in antioxidants, amino acids, high-quality proteins, iron and calcium, unsaturated fatty acids, and a variety of vitamins, including A, B2, B6, B8, B12, E, and K. Spirulina is shown to have antiviral, anti-inflammatory, and anticancer properties, as well as the ability to reduce blood lipid profile, blood sugar, body weight, hypertension, wound healing time and to enhance the growth of intestinal Lactobacillus. As a result, Spirulina is regarded as a functional food with therapeutic properties for a variety of ailments [12,13]. Recently, it was speculated that Spirulina could be associated with the modulation of the host immune system. In mice, Spirulina enhanced IL-1 and antibody production [14]. *Spirulina platensis* extract was regarded as the best algal source for prebiotics as it had a greater stimulatory effect on the growth of probiotic bacteria. Oligosaccharides in algal extracts function as prebiotic compounds for the stimulation of probiotic bacteria [15].

To the best of our knowledge, this is the first study evaluating the synergistic immunomodulatory and hypolipidemic effects of mixture of (*L. plantarum*, *L. casei*) and *Spirulina platensis* in an animal model of Swiss albino mice with diet-induced hypercholesterolemia.

## 2. Materials and Methods

### 2.1. Bacterial Strains

*Lactobacillus plantarum* p9 was isolated and identified according to Hoda et al. [16]. *Lactobacillus casei* ATCC 7469 and *Spirulina platensis* cyanobacteria were obtained from the Microbiology Lab at Genetic Engineering and Biotechnology Research Institute (GEBRI), Sadat University, Sadat City, Egypt.

### 2.2. Algae Cultivation

*Spirulina platensis* cyanobacteria were cultivated in Zarrouk's medium [17] at $25 \pm 2\ °C$, pH 10 with continuous illumination using cool white fluorescent tubes (2500 Lux) and was shaken by hand twice daily for 15 days. Cells were collected by filtration using filter paper with 8 mm pore size (screen-printing paper), washed with buffer solution (pH 7), diluted, and processed for further inoculation. A known volume of *S. platensis* sample was filtered through screen-printing paper and oven-dried at $60\ °C$ until a constant weight was reached.

### 2.3. Preparation of Cell Lysate

Cells ($1 \times 10^7$ cfu/mL) from *L. plantarum* and *L. casei* were harvested by centrifugation at 4 °C for 30 min (5000 rpm). The pellets were washed twice with 20 mM sodium phosphate buffer pH 7.4 and then re-suspended in it. The washed cell suspension was disrupted with an ultrasonic cell disrupter (Brandson 4 °C) and filtered (0.45 μm, Millipore, Burlington, MA, USA). Cells debris was removed by centrifugation ($10,000 \times g$ for 10 min), the lysate was collected, and its protein concentration was estimated by the Bradford method (Bio-Rad Laboratories, Hercules, CA, USA) [18].

### 2.4. Experimental Design for Mice Feeding with L. plantarum and L. casei and Spirulina platensis

2.4.1. Animals

Forty-five five- to six-week-old Swiss albino mice were obtained from National Organization of Drug Control and Research, Giza, Egypt. The mice were randomly housed in plastic cages and allowed to acclimatize for ten days before the experiment. All animal handling procedures, sample collection and disposal were carried out according to the regulation of Institutional Animal Care and Use Committee (IACUC), Faculty of Veterinary Medicine, University of Sadat City, Egypt, under approval number VUSC-001-3-16.

The mice were randomized into 9 groups that were treated as follows:

**G1:** Normal group (fed on regular diet).

**G2:** Fed on regular diet treated with 1 mg/kg BW of cell lysates of *L. plantarum* + *L. casei* ($1 \times 10^7$ cfu/mL) at 1:1 ratio for 8 consecutive weeks.

**G3:** Fed on regular diet treated with 1 mg/kg BW of *S. platensis* for 8 consecutive weeks.

**G4:** Fed on regular diet treated with 1 mg/kg BW of cell lysates of *L. plantarum* + *L. casei* ($1 \times 10^7$ cfu/mL) at 1:1 ratio + *S. platensins* for 8 consecutive weeks.

**G5:** Positive control (fed on high-cholesterol diet)**.**

**G6:** Fed on high-cholesterol diet treated with 1 mg/kg BW of cell lysates of *L. plantarum* + *L. casei* ($1 \times 10^7$ cfu/mL) at 1:1 ratio for 8 consecutive weeks.

**G7:** Fed on high-cholesterol diet treated with 1 mg/kg BW of *S. platensis* for 8 consecutive weeks.

**G8:** Fed on high-cholesterol diet treated with 1 mg/kg BW of cell lysates of *L. plantarum* + *L. casei* ($1 \times 10^7$ cfu/mL) at 1:1 ratio + *S. platensins* for 8 consecutive weeks.

**G9:** Fed on high-cholesterol diet treated with 1 mg/kg BW of standard rosuvastatin for 8 consecutive weeks.

The regular diet consists of wheat flour 22.5%, soybean powder 25%, essential fatty acids 0.6%, vitamins (A 0.6 mg/kg, D 1000 IU/kg, E 35 mg/kg, niacin 20 mg/kg, pantothenic acid 8 mg/kg, riboflavin 0.8 mg/1000 kcal, thiamin 4 mg/kg, B6 50 mg/kg, and B12 7 mg/kg of diet) and minerals (calcium 5 g/kg, phosphorus 4 g/kg, fluoride 1 mg/kg, iodine 0.15 mg/kg, chloride 5 mg/kg, iron 35 mg/kg, copper 5 mg/kg, magnesium 800 mg/kg, potassium 35 mg/kg, manganese 50 mg/kg, and sulfur 3 mg/kg of diet) [19]. The nutrition contents of the high-cholesterol diet were similar to those of the regular diet, except for the addition of (1000 μg/mice) cholesterol to the regular diet.

Body mass was measured at the beginning and at the end of the experiment.

2.4.2. Blood Sampling and Biochemical Assays

Animals of different groups were sacrificed under diethyl ether anesthesia. Blood samples were collected using ethylenediaminetetraacetic acid (EDTA). EDTA was used as an anticoagulant and centrifuged at 3000 rpm for 20 min. Sera were used for biochemical investigations. Triglycerides (TG) were determined according to the method of Fossati and Prencip [20]. Tricholesterol (TC) was determined according to the method of Deeg and Ziegenohrm [21]. HDL-C was determined in the plasma according to the method of Lopez et al. [22]. LDL-C was calculated according to the method of Friedewald et al. [23].

### 2.4.3. Histopathological Examination

At the end of the experiments, the mice were sacrificed under diethyl ether anesthesia, and spleen and heart were collected and fixed with 10% formalin. After 72 h of fixation, samples were dehydrated, embedded in paraffin wax, sectioned (6 μm), and then stained with hematoxylin and eosin (H&E). The stained tissues were then examined microscopically by a light microscope for general histopathological changes. Histological photos were taken by using a Leica EC3 digital camera [24].

### 2.4.4. Assessment of Complete Blood Count (CBC)

The mice were fasted overnight prior to sample collection. Diethyl ether was used to take blood samples from mice that were under general anesthesia. Venipuncture was used to draw blood samples, which were then placed in test tubes with anticoagulant (EDTA). The blood was transferred into the tube as soon as possible after collection, and the anticoagulant was mixed properly. Then, all samples were sent to the laboratory for the measurement of hematological parameters (white blood cells (WBC), platelet count, hemoglobin level (Hb), red blood cells (RBC), hematocrit level (HCT), mean corpuscular volume (MCV), mean corpuscular hemoglobin (MCH) and mean corpuscular hemoglobin concentration (MCHC). All samples were analyzed by VetScan HM2 $^{TM}$ Hematology System, Abaxis® (Union City, CA, USA).

### 2.4.5. Analysis of Myeloid Cells with Flow Cytometry

Erythrocytes were lysed in the blood samples with ACK in Cell Culture Grade Water. Cells were stained with anti-Ly6G mAb and anti-CD11b mAb (purchased from BD Biosciences, Sanjose, CA, USA) for 30 min on in ice and at dark. The stained cells were then washed twice with phosphate-buffered saline (PBS) and, finally, resuspended in flow cytometry staining buffer (FACS). The cells were then evaluated against a BD FACSCalibur TM and Partec (BD Biosciences, Sanjose, CA, USA) and analyzed using flow Jaw software, BD Biosciences [25].

### *2.5. Statistical Analysis*

Data are presented as the mean $\pm$ standard error (SE) and were subjected to statistical analysis using one-way analysis of variance (ANOVA) according to Snedcor and Cochran [26].

### 3. Results and Discussion

### *3.1. Effects of Lactobacillus and Spirulina Treatment on Body Weight*

Obesity is linked to a higher intake of high-calorie foods, which leads to increased fat storage and body mass [27]. Figure 1 demonstrates the effect of Lactobacilli and Spirulina on the body weight of mice fed on regular or high-cholesterol diets.

At 15 days and at 57 days, there was a significant increase in body weight of G5 ($p < 0.05$) compared to that of G1, of G2, G3 and G4. At 15 days, there was no significant change ($p < 0.05$) in body weight of G6, G7, G8 and G9 compared to G5, while the body weight of these groups significantly decreased ($p < 0.05$) compared to G5 at 57 days.

Crovesy et al. [28] reported that probiotics have the potential to help with weight and fat mass loss in overweight subjects. An increased short-chain fatty acids (SCFA) generation by probiotics influences appetite and energy homeostasis [29]. Some Bifidobacterium and Lactobacillus species have been found to produce beneficial conjugated linoleic acid (CLA). CLA improves energy metabolism and lipolysis, which has an effect on body weight [30]. Spirulina has a beneficial effect on reducing body fat, waist circumference, body mass index and appetite. The proposed mechanism of action of Spirulina is a reduction in macrophage infiltration into visceral fat, the prevention of hepatic fat accumulation, a reduction in oxidative stress, and improvement in insulin sensitivity and satiety [31].

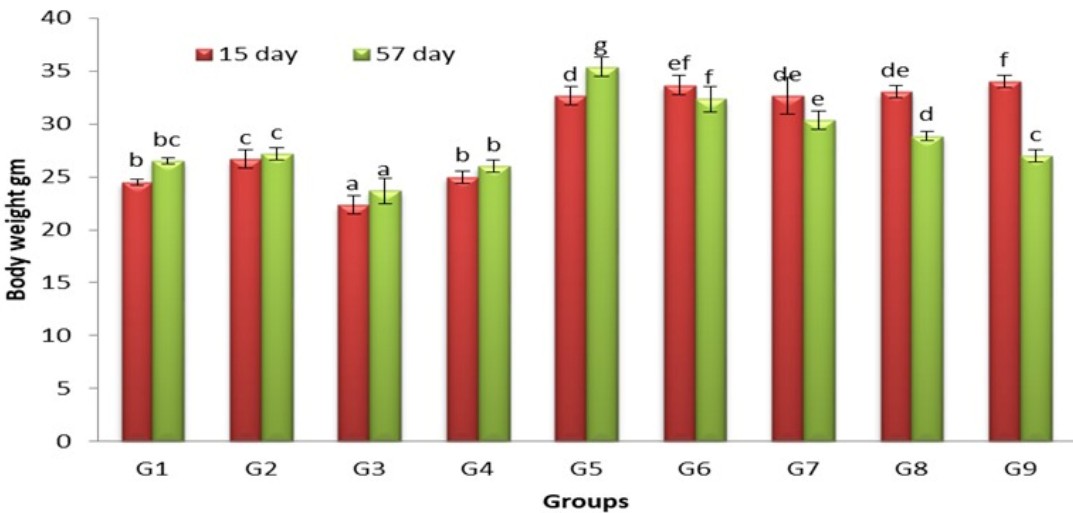

**Figure 1.** The influence of *L. plantarum* (Lp), *L. casei* (Lc), *S. platensis* (Sp) and their combination on body weight of albino mice at 15 and 57 days. Error bars represent SE of means. G1: regular diet (RD); G2: RD + Lp + Lc; G3: RD + Sp; G4: RD + Lp + Lc + Sp; G5: high-cholesterol diet (Ch); G6: Ch + Lp + Lc; G7: Ch + Sp; G8: Ch + Lp + Lc + Sp; G9: Ch+ rosuvastatin. Bars with different superscript are significantly different ($p < 0.05$).

### 3.2. Effects of Lactobacillus and Spirulina Treatment on Cholesterol Levels

Figure 2 demonstrates the effect of Lactobacilli and Spirulina on cholesterol levels of mice fed on regular or high-cholesterol diets.

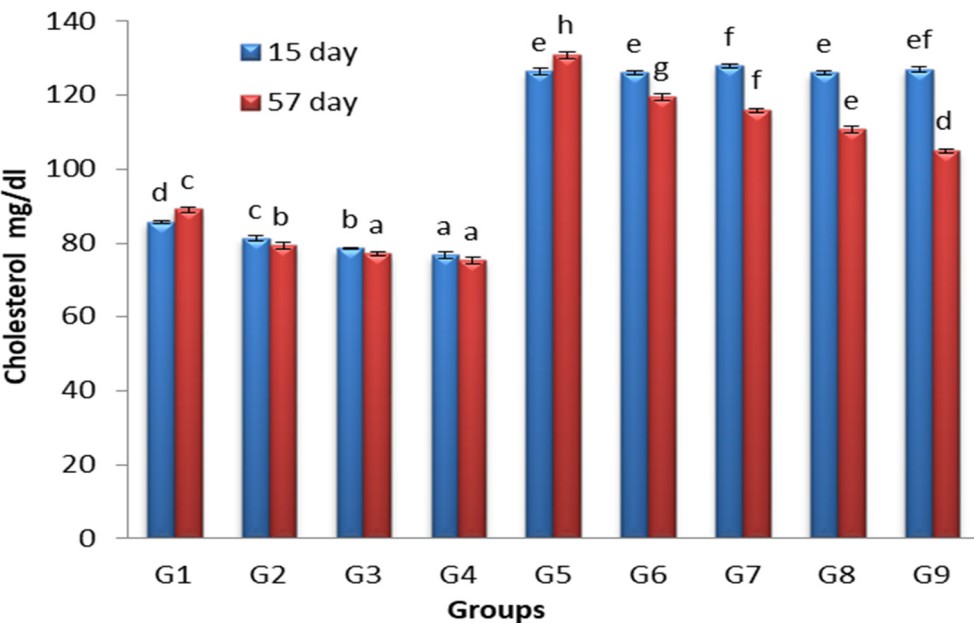

**Figure 2.** The influence of *L. plantarum* (Lp), *L. casei* (Lc), *S. platensis* (Sp) and their combination on total cholesterol (TC) level in albino mice (mg/dL) at 15 and 57 days. Error bars represent SE of mean. G1: Regular diet (RD); G2: RD + Lp + Lc; G3: RD + Sp; G4: RD + Lp + Lc + Sp; G5: high-cholesterol diet (Ch); G6: Ch + Lp + Lc; G7: Ch + Sp; G8: Ch + Lp + Lc + Sp; G9: Ch+ rosuvastatin. Bars with different superscript are significantly different ($p < 0.05$).

At 15 days and 57 days, there was significant increase ($p < 0.05$) in the TC level of G5 compared to G1, G2, G3 and G4. At 15 days, there was no significant change ($p < 0.05$) in TC level of G6, G7, G8 and G9 compared to G5, while the TC level of G6, G7, G8 and G9 significantly decreased ($p < 0.05$) compared to G5 at 57 days.

Wang et al. [32] stated that, in rats fed on a high-fat diet, Lactobacilli reduced cholesterol levels. The increase in fecal bile acid excretion by lactic acid bacteria may decrease blood cholesterol [33,34]. The deconjugation of bile and binding to bile acid by lactic acid bacteria in the small intestine has been postulated as a possible explanation for the increased fecal excretion of bile acids [35]. Because free bile acids are expelled more quickly than conjugated bile acids, the deconjugation of bile acids in the small intestine may result in increased bile acid excretion from the intestinal tract [36]. Jeun et al. [37] stated that the oral administration of *Lactobacillus plantarum* KCTC3928 to mice increased fecal bile–acid excretion, hepatic bile–acid production, and expression of 7-alpha-hydroxylase (CYP7A1), the major enzyme in cholesterol catabolism and bile–acid synthesis. According to Dvir et al. [38], Spirulina carbohydrates and dietary fibers lower cholesterol through increasing the size of the bile acid pool and fecal steroid excretion.

### 3.3. Effects of Lactobacillus and Spirulina Treatment on TG Level

Hypertriglyceridemia is a common risk factor for CVD, and it is becoming more widespread as the obesity and insulin resistance epidemics spread. High TG levels are indicators of atherogenic lipoproteins of various sorts [39]. Figure 3 demonstrates the effect of Lactobacilli and Spirulina on TG levels of mice fed on regular or high-cholesterol diets.

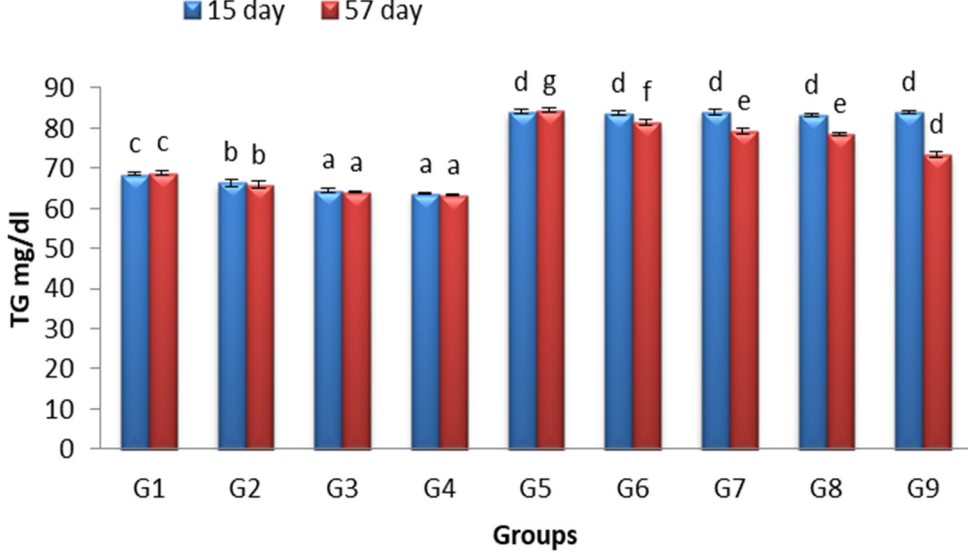

**Figure 3.** The influence of *L. plantarum* (Lp), *L. casei* (Lc), *S. platensis* (Sp) and their combination on triglyceride (TG) level in albino mice (mg/dl) at 15 days and at the end of experiment. Error bars represent SE of mean. G1: regular diet (RD); G2: RD + Lp + Lc; G3: RD + Sp; G4: RD + Lp + Lc + Sp; G5: high-cholesterol diet (Ch); G6: Ch + Lp + Lc; G7: Ch + Sp; G8: Ch + Lp + Lc + Sp; G9: Ch+ rosuvastatin. Bars with different superscript are significantly different ($p < 0.05$).

At 15 days and 57 days, there was significant increase ($p < 0.05$) in the TG level of G5 compared to G1, G2, G3 and G4. At 15 days, there was no significant change ($p < 0.05$) in TG level of G6, G7, G8 and G9 compared to G5, while TG level of G6, G7, G8 and G9 decreased significantly ($p < 0.05$) compared to G5 at 57 days.

Choi et al. [40] stated that *L. plantarum* KY1032 and *L. curvatus* HY7601 lowered triglycerides in hypertriglyceridemic rats by upregulating the expression of ApoAV, PPARα, and FXR. Ahn et al. [41] stated that the triglyceride-lowering effects of probiotic supplementation were related to elevated apoA-V. Mazokopakis et al. [42] stated that Spirulina had powerful hypolipidemic effects, especially on the triglyceride concentration in dyslipidaemic Cretan outpatients. Han et al. [43] reported that a glycolipid derived from Spirulina called glycolipid H-b2 inhibited pancreatic lipase activity in a dose-dependent manner and reduced postprandial TG levels. This action is thought to be secondary to the activation

of the AMP-activated protein kinase signaling pathway, which downregulates the expression of lipid synthesizing genes, such as sterol regulatory element-binding transcription factor-1c, 3-hydroxy-3-methyl glutaryl coenzyme A reductase, and acetyl CoA carboxylase, lowering TG levels and inhibiting fatty acid synthesis [31].

### 3.4. Effects of Lactobacillus and Spirulina Treatment on High-Density Lipoprotein Level (HDL)

The ability of HDL to collect and return excess cholesterol from peripheral tissues to the liver, and thus the ability of its function in preventing atherosclerosis, myocardial infarction, transient ischemic attack, and stroke, has attracted researchers' curiosity [44]. Figure 4 demonstrates the effect of Lactobacilli and Spirulina on HDL levels of mice fed on regular or high-cholesterol diets.

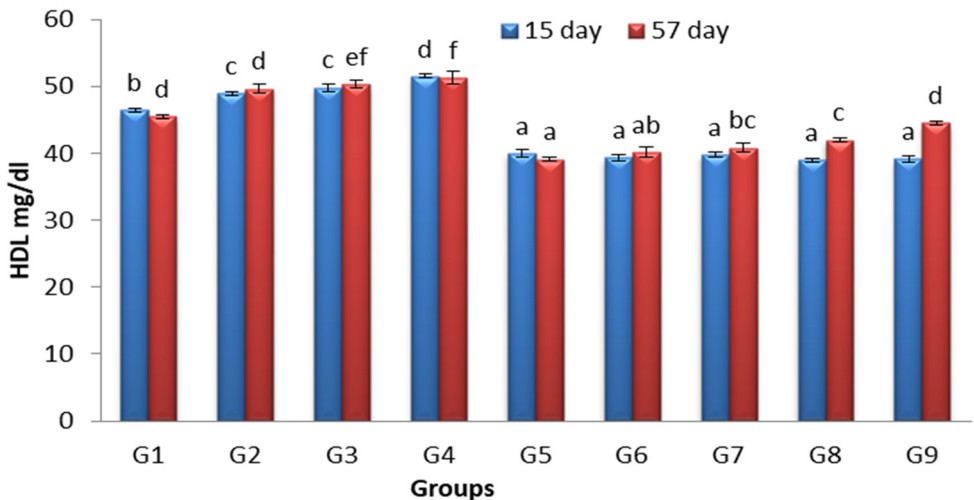

**Figure 4.** The influence of *L. plantarum* (Lp), *L. casei* (Lc), *S. platensis* (Sp) and their combination on high-density lipoprotein (HDL-C) level in albino mice (mg/dL) at 15 and 57 days. Error bars represent SE of mean. G1: regular diet (RD); G2: RD + Lp + Lc; G3: RD + Sp; G4: RD + Lp + Lc + Sp; G5: high-cholesterol diet (Ch); G6: Ch + Lp + Lc; G7: Ch + Sp; G8: Ch + Lp + Lc + Sp; G9: Ch+ rosuvastatin. Bars with different superscript are significantly different ($p < 0.05$).

At 15 days and 57 days, there was a significant decrease ($p < 0.05$) in HDL-C level of G5 compared to G1, G2, G3 and G4. At 15 days, there was no significant change ($p < 0.05$) in HDL-C level of G6, G7, G8 and G9 compared to G5, while the HDL-C level of G6, G7, G8 and G9 significantly increased ($p < 0.05$) compared to G5 at 57 days.

Chaiyasut et al. [45] stated that *L. paracasei* HII01 significantly increased HDL-C in hypercholesterolemia patients. Li et al. [46] found that Spirulina given for 8 weeks increased HDL-C in rats fed a high-fat diet. In the core of HDL-C, cholesterol is transferred as cholesteryl esters. According to Ooi and Liong [47], probiotics generated a hypocholesterolemic impact by changing cholesteryl esters and lipoprotein transporter pathways. Nagoka et al. [48] discovered that a new protein, C-phycocyanin, produced from Spirulina, which contains a substantial quantity of cystine and is responsible for increasing HDL-C.

### 3.5. Effects of Lactobacillus and Spirulina Treatment on Low-Density Lipoprotein Level (LDL)

LDL cholesterol and Apolipoprotein B (ApoB), the major structural protein of LDL, are both linked to an increased risk of atherosclerotic cardiovascular events (Sniderman et al.) [49]. Figure 5 demonstrates the effect of Lactobacilli and Spirulina on LDL levels of mice fed on regular or high-cholesterol diets.

At 15 days and 57 days, there was a significant increase ($p < 0.05$) in LDL-C level of G5 compared to G1, G2, G3 and G4. At 15 days, there was no significant change ($p < 0.05$) in the LDL-C levels of G6, G7, G8 and G9 compared to G5, while the LDL-C levels of G6, G7, G8 and G9 significantly decreased ($p < 0.05$) compared to G5 at 57 days.

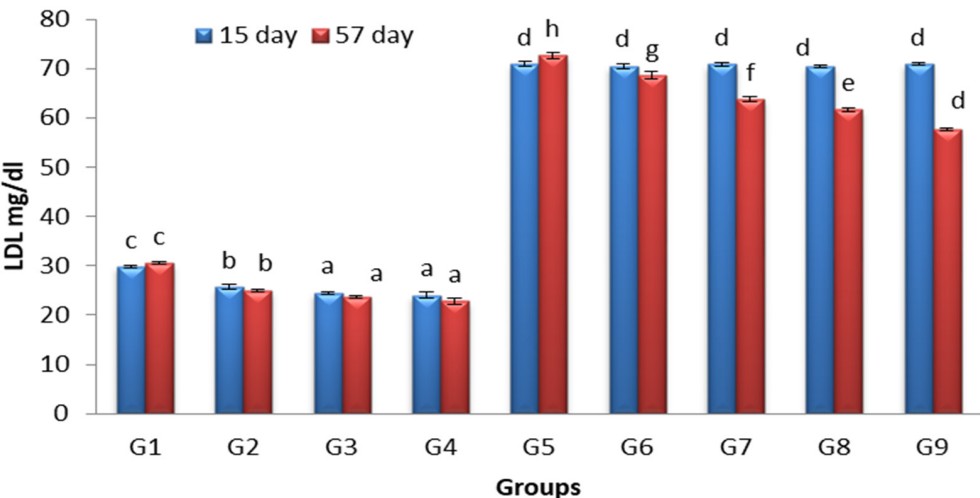

**Figure 5.** The influence of *L. plantarum*, *L. casei*, *S. platens* and their combination on low-density lipoprotein cholesterol (LDL-C) levels in albino mice (mg/dL) at 15 and 57 days. Error bars represent SE of mean. G1: regular diet (RD); G2: RD + Lp + Lc; G3: RD + Sp; G4: RD + Lp + Lc + Sp; G5: high-cholesterol diet (Ch); G6: Ch + Lp + Lc; G7: Ch + Sp; G8: Ch + Lp + Lc + Sp; G9: Ch+ rosuvastatin. Bars with different superscript are significantly different $p < 0.05$.

Wu et al. [50] stated that consuming probiotic Lactobacillus, especially *L. reuteri* and *L. plantarm*, could significantly reduce LDL-C. The beneficial effects of probiotics are suggested to be due to their ability to regulate lipid metabolism by modulating the expression of certain genes that are involved in the lipid biosynthesis pathway [51]. Cheong et al. [52] stated that Spirulina decreased LDL-C in rabbits fed on high-fat diets.

### 3.6. Complete Blood Count

Table 1 demonstrates the effect of Lactobacilli and Spirulina on the blood count of mice fed on a regular or high-cholesterol diet.

Treatments with Lactobacillus (G2) or Spirulina (G3) or by their combination (G4) induced a significant increase in the Hb level, MCHC level and HCT level in mice fed on regular diet (G1) ($p < 0.05$). There was also a significant increase in Hb level, MCHC level and HCT level in mice fed on high-cholesterol diet (G5). Treatments with Lactobacillus (G6) or Spirulina (G7) or by their combination (G8) did not induce a significant effect on these parameters ($p < 0.05$) in G5, while treatment with standard rosuvastatin (G9) induced a significant decrease in these parameters in G5 ($p < 0.05$). All treatments did not induce significant changes in MCH, RBC count and MCV red cell indices ($p < 0.05$).

Treatments with Lactobacillus (G2) or Spirulina (G3) or by their combination (G4) induced a significant increase in the platelet count and WBC number of mice fed on a regular diet $p < 0.05$. The largest increase in the previous parameters was in mice fed on a high-cholesterol diet (G5) $p < 0.05$. The treatment with Lactobacillus (G6) or Spirulina (G7) or by their combination (G8) or by standard rosuvastatin (G9) resulted in a significant decrease in the previous parameters in mice fed on high-cholesterol diets (G5) $p < 0.05$.

The oral supplementations with *Lactobacillus plantarum* increased the levels of Hb, PCV and RBC in Wistar albino rats [53]. *Spirulina platensis* improved hematological parameters in both diabetic and non-diabetic rats [54]. Dias et al. [55] stated that the addition of probiotics to the diet of caged matrinxã increased HCT, RBC. Korčok et al. [56] linked this finding to the activation of the hematopoietic organs, as well as the indirect influence of some lactic acid bacteria, such as Lactobacilli, on enhancing the bioavailability of dietary iron through a variety of processes, including lowering intestinal pH. Pacheco et al. [57] stated that hemolytic anemia may be a rare side effect of Atorvastatin and Lovastatin.

**Table 1.** The influence of *L. plantarum*, *L. casei*, *S. platensis* and in their combination on complete blood count of albino mice at 15 and 57 days.

| Parameter | Days | G1 | G2 | G3 | G4 | G5 | G6 | G7 | G8 | G9 |
|---|---|---|---|---|---|---|---|---|---|---|
| Hb (g/dL) | 15 | 8.5 ± 0.288 a | 9.6 ± 0.23 ab | 10.3 ± 0.05 bc | 10.7 ± 0.12 bcd | 11.5 ± 0.12 cde | 11.3 ± 0.17 cde | 12 ± 0.28 e | 11.4 ± 0.29 cde | 11.7 ± 0.12 de |
| | 57 | 8.9 ± 0.284 a | 10.2 ± 0.12 abc | 10.9 ± 0.1 bcd | 11.2 ± 0.14 cd | 12.1 ± 0.12 d | 11.2 ± 0.18 cd | 11.7 ± 0.23 d | 11.3 ± 0.26 cd | 9.6 ± 0.19 ab |
| RBC (M/μL) | 15 | 3 ± 0.12 a | 3.4 ± 0.06 ab | 3.6 ± 0.05 ab | 3.7 ± 0.06 ab | 3.9 ± 0.06 ab | 3.8 ± 0.05 b | 4.2 ± 0.11 ab | 3.8 ± 0.12 ab | 4.1 ± 0.06 ab |
| | 57 | 3.2 ± 0.12 a | 3.5 ± 0.06 a | 3.7 ± 0.03 a | 3.9 ± 0.06 a | 4.16 ± 0.08 a | 3.8 ± 0.08 a | 4.03 ± 0.08 a | 3.7 ± 0.08 a | 3.4 ± 0.03 a |
| HCT (%) | 15 | 26 ± 0.01 a | 29.4 ± 0.01 b | 31.2 ± 0.01 c | 32.1 ± 0.01 cd | 33.8 ± 0.01 ef | 32.9 ± 0.01 cd | 36.4 ± 0.01 g | 33.5 ± 0.01 de | 35.5 ± 0.01 fg |
| | 57 | 27 ± 0.01 a | 30.3 ± 0.01 b | 32.6 ± 0.01 c | 33.8 ± 0.01 de | 35.7 ± 0.01 f | 33.1 ± 0.01 cde | 34.9 ± 0.01 ef | 32.6 ± 0.01 cd | 29.7 ± 0.01 b |
| MCH (pg) | 15 | 28.3 ± 0.15 a | 28.3 ± 0.18 a | 28.6 ± 0.32 a | 28.9 ± 0.15 a | 29.5 ± 0.12 a | 29.7 ± 0.03 a | 28.6 ± 0.12 a | 29.5 ± 0.13 a | 28.5 ± 0.15 a |
| | 57 | 28.1 ± 0.4 a | 29.1 ± 0.15 ab | 28.9 ± 0.01 ab | 28.8 ± 0.05 ab | 28.9 ± 0.27 ab | 29.3 ± 0.27 ab | 29.1 ± 0.08 ab | 30 ± 0.26 b | 28 ± 0.29 a |
| MCHC (g/dL) | 15 | 32.7 ± 0.12 ab | 32.5 ± 0.23 a | 33 ± 0.37 ab | 33.3 ± 0.17 bc | 33.9 ± 0.14 c | 34.2 ± 0.03 c | 32.9 ± 0.08 ab | 34.1 ± 0.15 c | 32.9 ± 0.12 ab |
| | 57 | 32.3 ± 0.49 a | 33.6 ± 0.17 b | 33.4 ± 0.03 b | 33.2 ± 0.09 b | 33.6 ± 0.3 b | 33.8 ± 0.32 b | 33.6 ± 0.09 b | 34.6 ± 0.26 c | 32.4 ± 0.32 a |
| MCV (fL) | 15 | 86.7 ± 0.06 a | 86.7 ± 0.05 a | 86.7 ± 0.05 a | 86.7 ± 0.05a | 86.7 ± 0.03 a | 86.7 ± 0.03 a | 86.6 ± 0.04 a | 86.7 ± 0.08 a | 86.7 ± 0.05 a |
| | 57 | 86.7 ± 0.06 a | 86.7 ± 0.05 a | 86.6 ± 0.06 a | 86.6 ± 0.05 a | 85.98 ± 0.04 a | 86.5 ± 0.03 a | 86.6 ± 0.05 a | 86.6 ± 0.02 a | 86.6 ± 0.08 a |
| WBC (×109/L) | 15 | 5 ± 0.06 a | 5.4 ± 0.12 b | 5.8 ± 0.12 c | 6.2 ± 0.05 d | 11.8 ± 0.08 e | 11.6 ± 0.1 e | 12.1 ± 0.1 f | 11.8 ± 0.1 e | 12.1 ± 0.1 f |
| | 57 | 4.9 ± 0.08 a | 5.6 ± 0.12 b | 6 ± 0.12 c | 6.3 ± 0.06 d | 12 ± 0.2 i | 11 ± 0.1 h | 10.6 ± 0.1 g | 9.2 ± 0.1 f | 6.7 ± 0.1 e |
| Platelets (×109/L) | 15 | 205 ± 0.9 a | 217 ± 0.98 b | 248 ± 0.88 c | 257.66 ± 0.9 c | 368 ± 0.8 d | 367 ± 0.9 d | 370 ± 0.9 d | 368.66 ± 0.8 d | 369.66 ± 0.9 d |
| | 57 | 204 ± 0.8 a | 232 ± 0.7a | 253.66 ± 0.8 ab | 257.6 ± 0.89 ab | 370 ± 0.8 c | 335 ± 0.9 bc | 309 ± 0.9 a | 290.66 ± 0.8 abc | 216.33 ± 0.99 a |

G1: regular diet (RD); G2: RD + Lp + Lc; G3: RD + Sp; G4: RD + Lp + Lc + Sp; G5: high-cholesterol diet (Ch); G6: Ch + Lp + Lc; G7: Ch + Sp; G8: Ch + Lp + Lc + Sp; G9: Ch+ rosuvastatin. Values are means and standard errors for 3 mice per treatment. Data in the rows with different superscripts are significantly different ($p < 0.05$).

### 3.7. Analysis of Myeloid Cells with Flow Cytometry

Flow cytometry is an important method that allows for the delineation of specific cell components of immune responses and disease states [58]. Monocytes and macrophages are innate immune system cells that protect against pathogen invasion by producing cytotoxic chemicals, such as reactive oxygen species (ROS), and secreting proinflammatory cytokines, such as TNF- and IL-8 [59].

Treatments with Lactobacillus (G2) or Spirulina (G3) or by their combination (G4) caused a significant increase in mature and immature neutrophils as well as monocytes in mice fed a regular diet (G1) $p < 0.05$.

Most probiotics stimulate innate immune defenses (phagocytosis, pro-inflammatory cytokines) and act positively for the duration of infectious episodes, in particular neutrophils, which play a key role in the immune response [60]. Watanuki et al. [14] stated that dietary Spirulina enhanced responses of phagocytic activity responses, interleukin (IL)-1b expression, and tumor necrosis factor (TNF)-$\alpha$ genes in carp. Spirulina also boosted immunity by different mechanisms, including increasing antibody production by B cells, inducing the activation of innate immune cells, such as monocytes and macrophages, augmenting interferon production by natural killer cells [61]. The immunostimulant effects of Spirulina were mainly mediated by its polysaccharide [62].

The largest increase in mature, immature neutrophils and monocytes was recorded in mice fed a high-cholesterol diet (G5). Hypercholesterolemia increases circulating monocyte counts and renders these cells more prone to emigration into atherosclerotic lesions [63].

There was significant decrease in mature and immature neutrophils as well as monocytes in treatments with Lactobacillus (G6) or Spirulina (G7), by their combination (G8), or by standard rosuvastatin (G9) compared to G5 $p < 0.05$ (Table 2).

The improvement in lipid profile after treatments by Lactobacillus and Spirulina explains the decreased migration of neutrophils and monocytes compared to positive controls (G5). *Lactobacillus plantarum* was found to diminish pulmonary inflammation in *Klebsiella pneumoniae*-infected mice, as evidenced by a decrease in macrophages and neutrophils, as well as pro-inflammatory cytokines (KC, IL-6, and TNF-) and the blockage of NF-B activation through an interaction with TLR [64]. Cristofori et al. [65] stated that probiotic therapy may minimize the development of inflammatory biomarkers and the blunt unnecessary activation of the immune system. This therapy could be used to achieve an immune modulation without the possible risks related to living microorganisms, such as infections in immune-deficient patients. Beneficial immune-modulatory effects are elicited across several molecules, which include microbial cell walls, peptidoglycan, and exopolysaccharides, through interactions with specific host cell receptors (i.e., toll-like receptor (TLR)-2 and TLR-4) [66].

**Table 2.** The influence of *L. plantarum*, *L. casei*, *S. platensis* and in their combination on myeloid cells in blood of albino mice.

| | G1 | G2 | G3 | G4 | G5 | G6 | G7 | G8 | G9 |
|---|---|---|---|---|---|---|---|---|---|
| CD11b − Ly6G+ | 3 ± 0.1 [a] | 3.84 ± 0.1 [b] | 4.74 ± 0.2 [c] | 5.8 ± 0.1 [d] | 14.2 ± 0.2 [i] | 11.9 ± 0.4 [h] | 10.5 ± 0.2 [g] | 9.66 ± 0.1 [f] | 7.5 ± 0.2 [e] |
| CD11b + Ly6G+ | 5.1 ± 0.1 [a] | 5.9 ± 0.1 [b] | 6.35 ± 0.1 [c] | 6.7 ± 0.05 [d] | 45.2 ± 0.6 [i] | 42.8 ± 0.9 [h] | 41.5 ± 0.2 [g] | 37.26 ± 0.2 [f] | 24.2 ± 0.1 [e] |
| CD11b + Ly6G− | 0.7 ± 0.04 [a] | 0.9 ± 0.02 [b] | 1.72 ± 0.04 [c] | 3 ± 0.11 [d] | 25.8 ± 0.6 [i] | 23.8 ± 0.17 [h] | 21.8 ± 0.3 [g] | 20.5 ± 0.2 [f] | 12.9 ± 0.1 [e] |

The phenotypes of immune cells: CD11b + Ly6G+ immature neutrophil; CD11b − Ly6G+ mature neutrophil; CD11b + Ly6G monocyte. Error bars represent SE of mean. G1: Regular diet (RD); G2: RD + Lp + Lc; G3: RD + Sp; G4: RD + Lp + Lc + Sp; G5: high-cholesterol diet (Ch); G6: Ch + Lp + Lc; G7: Ch + Sp; G8: Ch + Lp + Lc + Sp; G9: Ch+ rosuvastatin. Values are means and standard errors for 3 mice per treatment. Data in the rows with different superscripts are significantly different ($p < 0.05$).

### 3.8. Histopathological Examination

3.8.1. Effects on Heart

The microscopic examination of cardiac tissue of mice fed on high-cholesterol diets (G5) showed moderate pathological alterations, where sub pericardium hemorrhagic areas, together with interstitial inflammatory aggregates, were seen. Areas of cardiomyocytes displayed degenerative changes in the form of an eosinophilic cytoplasm and pyknotic nuclei (Figure 6a). The cardiac tissue of mice fed a high-cholesterol diet treated with Lactobacillus (G6) displayed intact cardiomyocytes and few interstitial hemorrhagic areas in (Figure 6b). The histological examination of cardiac tissues of mice fed on a high-cholesterol diet treated with Spirulina (G7) showed an ameliorating effect of *Spirulina* on cardiac tissues, where wide areas of intact cardiomyocytes were observed; however, lipid-laden macrophage aggregates, together with an interstitial edema and congested vasculature, could be seen in (Figure 6c). Cardiac tissues of mice fed on a high-cholesterol diet treated with Lactobacillus and Spirulina (G8) displayed the most curative results, as it showed intact cardiomyocytes (Figure 6d). Cardiac tissue of mice fed on high-cholesterol diet treated with rosuvastatin (G9) showed marked reduction in inflammation and congestion of vasculature (Figure 6e).

Similar findings were observed by AL-Aameli et al. [67], who stated that the cardiac muscle layer in rats fed on regular diets was striated and arranged in a linear pattern that branches and anatomizes in a specific model, giving the appearance of a sheet, while cholesterol-fed rats showed signs of heart tissue damage such as myofibrillar loss, blood vessel congestion, vacuolation, and lipid aggregation. Sadeghzadeh et al. [68] stated that pretreatment with *Lactobacillus casei* considerably reduced myocardial necrosis, edema, and the infiltration of inflammatory cells in hyperlipidemic rat models.

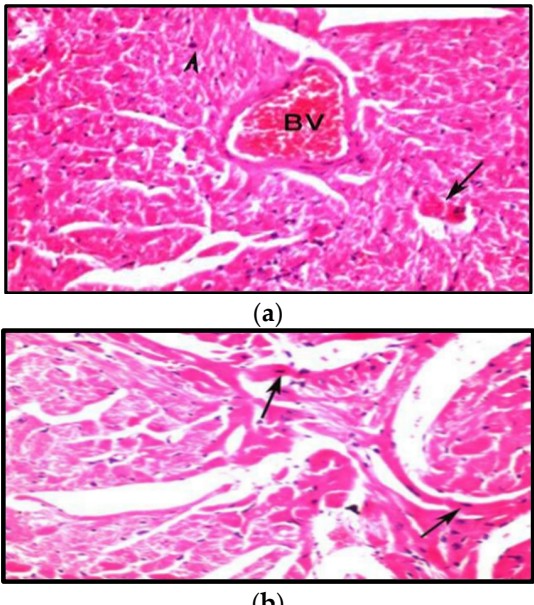

(**a**)

(**b**)

**Figure 6.** *Cont.*

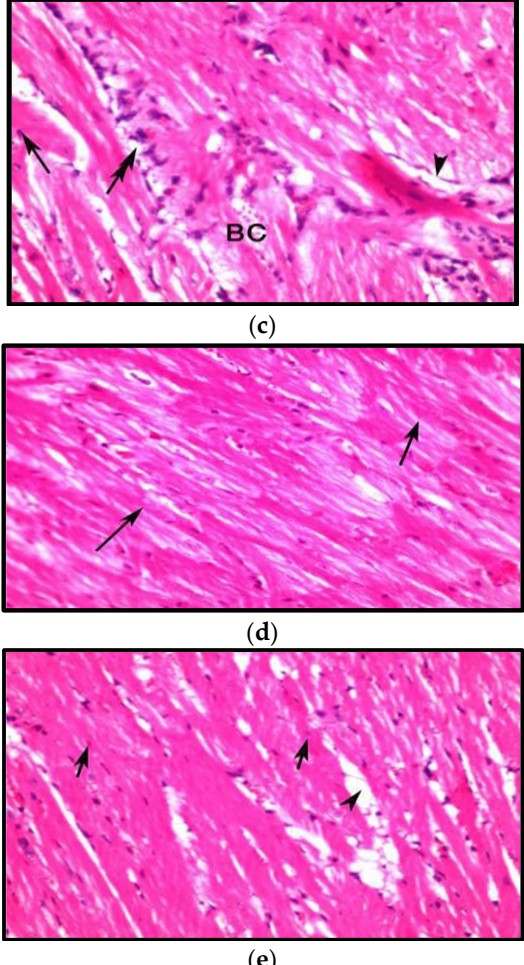

**Figure 6.** Histology of heart section stained with hematoxylin and eosin (original magnification ×200) from each group: G5: high-cholesterol diet (Ch); G6: Ch + Lp + Lc; G7: Ch + Sp; G8: Ch + Lp + Sp; G9: Ch + rosuvastatin. (**a**) G5: Section in cardiac tissue showing dilated congested blood vessels, many cardiomyocytes with deeply stained cytoplasm, and pyknotic nuclei (arrow head) atropheid cardiomycetes (H&E. 400×). (**b**) G6: Section in cardiac tissue showing intact cardiomyocytes and few interstitial hemorrhagic areas with pyknotic nuclei (arrowhead) (H&E. 400×). (**c**) G7, Section in cardiac tissue showing pyknotic nuclei and deeply stained cytoplasm (arrow), inflammatory cells infiltrate (double arrow), dilated blood capillaries (BC), edema (arrowhead) (H&E. 400×). (**d**) G8: Section in cardiac tissue showing intact cardiomyocytes (arrow) (H&E. 400×). (**e**) G9: Section in cardiac tissue showing intact cardiomyocytes (arrow) with vesicular nucleus, mild edema in interstitial spaces (arrowhead) (H&E. 400×).

### 3.8.2. Effect on the Spleen

The microscopic examination of splenic tissue of mice fed on high-cholesterol diets (G5) showed thick capsules with thick trabecular bundles of fibrosis, edema, and thick connective tissue, as well as an absence of demarcation between red and white pulp in 50% of the animals with degenerative changes in lymphocytes. Some red pulp showed atrophy, but the white pulp contained deposited eosinophilic material (Figure 7a). In mice fed on high-cholesterol diets treated with Lactobacillus (G6), there was a marked improvement in the white and red pulp demarcation. However, an interstitial hemorrhage with brown hemosiderin pigments and mild megakaryocytes occurred in 50% of cases, where very dilated sinusoids were observed, and a compressed central vein with a thick wall and thick trabecular were present (Figure 7b). Splenic tissue of mice fed on high-cholesterol diet treated with Spirulina (G7) showed a normal appearance of the white and red pulp with

thick trabecular passing through it (Figure 7c). Splenic tissue of mice fed on high-cholesterol diet treated with Lactobacillus and Spirulina (G8) showed a normal appearance of white pulp (wp), red pulp (Rp) and the splenic trabecula (Figure 7d). In splenic tissue of mice fed on high cholesterol diet treated with rosuvastatin (G9), there was a demarcation between white and red pulp, but the reduction in the white pulp size was still observed in (Figure 7e).

Naghashpour et al. [69] stated that the microscopic examination of splenic tissue of hypercholesteremic patients revealed characteristic large histiocytes containing numerous granules of various sizes and shapes. Some granules were electron-dense with a homogeneous appearance. Shokryazdan et al. [70] stated that the administration of *L. buchneri* and *L. fermentum* HM3 to rats demonstrated a normal histological structure of spleen, comprising of sinusoid (S), lymphocytes (L) and red blood cells (R)).

### 3.8.3. Effects on Aorta

Microscopic examination of aorta in mice fed on high-cholesterol diet (G5) showed fibrin deposits on surface of the endothelium layer of tunica intima and a marked reduction in the thickness of tunica media (Figure 8a). Aorta of mice fed on high-cholesterol diet treated with Lactobacillus (G6) indicated an irregular endothelium layer and the vacuolation of tunica intima, together with pyknotic nuclei of smooth muscle and a little reduction in thickness of tunica media in (Figure 8b). The aorta of mice fed on high-cholesterol diets treated with Spirulina (G7) showed irregular endothelial layer in the tunica intima and the pyknotic nuclei of smooth muscle; it had normal thickness of tunica media in (Figure 8c). Aorta of mice fed on high-cholesterol diet treated with Lactobacillus and Spirulina (G8) showed a normal appearance of the tunica intima, smooth muscle, and elastic lamellae in tunica media (Figure 8d). The aorta of mice fed on high cholesterol diet treated with rosuvastatin (G9) showed intact endothelium layer, smooth muscle and elastic lamellae in tunica media (Figure 8e).

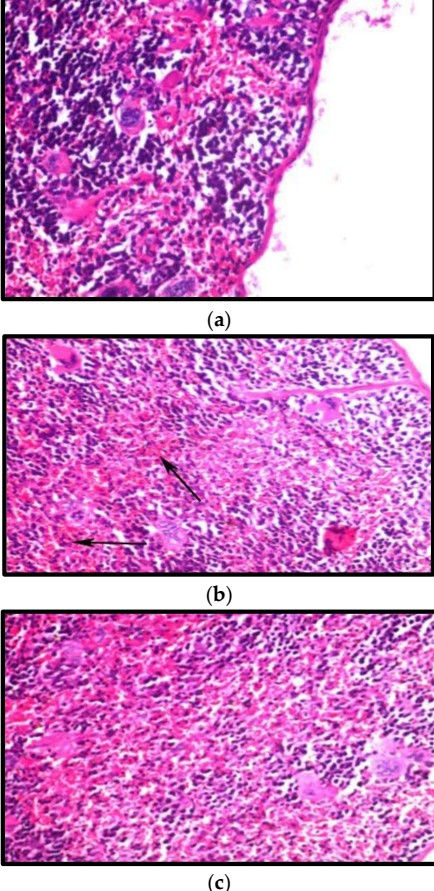

(a)

(b)

(c)

**Figure 7.** *Cont.*

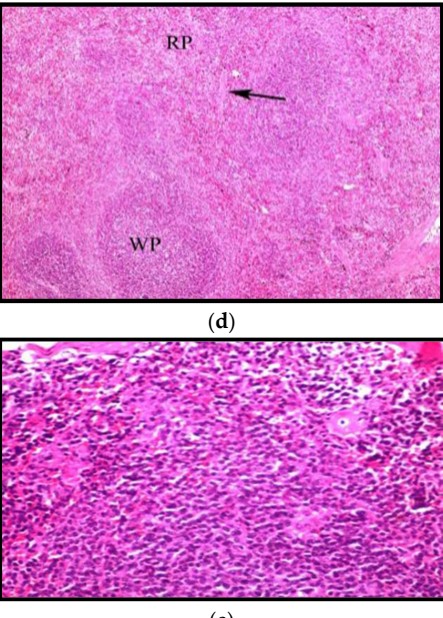

(**d**)

(**e**)

**Figure 7.** Histology of spleen section stained with hematoxylin and eosin (original magnification ×200) from each group: G5: high-cholesterol diet (Ch); G6: Ch + Lp + Lc; G7: Ch + Sp; G8: Ch + Lp + Sp; G9: Ch + rosuvastatin. (**a**) G5: Section in splenic tissue showing degenerative changes in the lymphocytes of the red pulp (R) and megakaryocytes (arrow) (H&E. 400×). (**b**) G6: Section in splenic tissue showing interstitial hemorrhage in the splenic sinusoid with brown pigments (arrow) (H&E. 400×). (**c**) G7: Section in splenic tissue showing normal appearance of white and red pulp elements (H&E. 400×). (**d**) G8: Section in splenic tissue showing normal appearance of white pulp (wp), red pulp (Rp) and the splenic trabecula (H&E. 400×). (**e**) G9: Section in splenic tissue showing a reduction in the white pulp elements (H&E. 400×).

According to AL-Aameli et al. [67], the aorta of the cholesterol-fed group of rats showed: multifocal collapse, necrosis, the disorientation of smooth muscle cells, an unequal wall of the aorta with increased wall width, damage to normal corrugation, intermittent endothelium in the tunica intima, and a hemorrhage in perivascular tissue (T. adventitia), with a vacuolation in the cells of the tunica media, accumulation of adipose tissue in the tunica adventitia, a minor increase in the thickness of the aorta wall, and partial stenosis in the lumen of the artery. Meanwhile, the aorta of the rats in the normal control group had a normal histological architecture and aortic thickness. The smooth intima of the typical main artery was organized and contained endothelial cells, a normal contour, and a normal endothelial corrugation of the intima. Inflammatory cell infiltration into the sub-endothelial layer was not observed. Furthermore, the adventitia lacked adipocytes. T media covered a considerable number of elastic fibers, as well as a substantial number of smooth muscle cells with distinct nuclei. The tunica adventitia of the aorta wall of normal rats showed a normal quantity of collagen fibers and connective tissue. Nabi et al. [71] stated that Lactobacillus treatment decreased the intimal layer and foam cells of the aortas of hyperlipidemic rats.

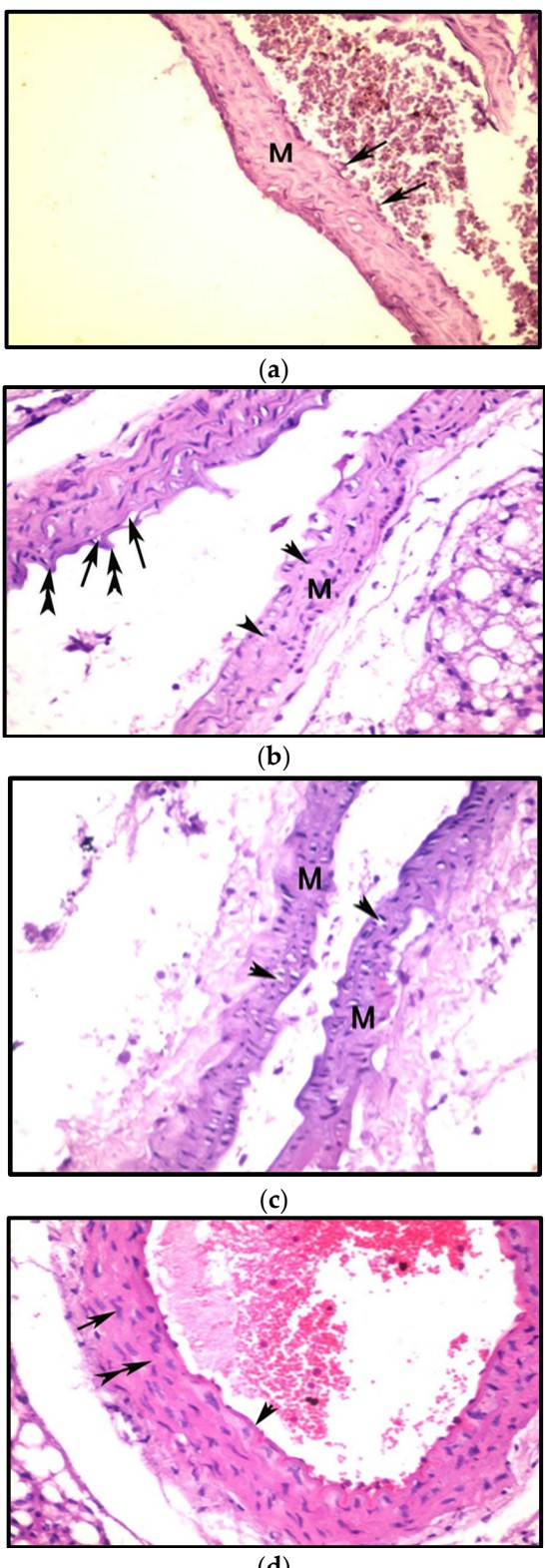

**Figure 8.** *Cont.*

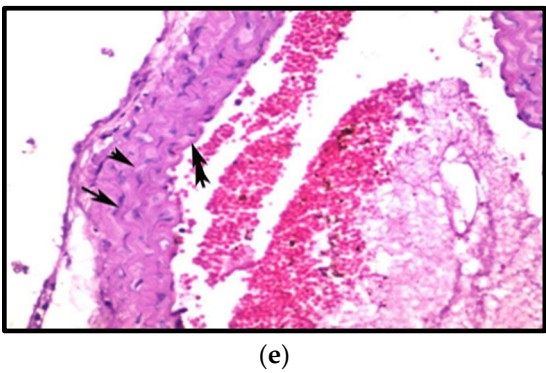

(**e**)

**Figure 8.** Histology of aortic section stained with hematoxylin and eosin (original magnification ×200) from each group: G5: high-cholesterol diet (Ch); G6: Ch + Lp + Lc; G7: Ch + Sp; G8: Ch + Lp + Sp; G9: Ch + rosuvastatin. (**a**) G5: Section in aorta showing fibrin deposits on surface of endothelium layer of tunica intima and marked reduction in thickness of tunica media (H&E. 400×). (**b**) G6: Section in aorta showing irregular endothelium layer, and vacuolation of tunica intima together with pyknotic nuclei of smooth muscle, as well as little reduction in thickness of tunica media (H&E. 400×). (**c**) G7: Section in aorta showing multiply irregular endothelial layers in tunica intima, pyknotic nuclei of smooth muscle and normal thickness of tunica media (H&E. 400×). (**d**) G8: Section in aorta showing normal appearance of tunica intima (arrow), smooth muscle (arrow) and elastic lamellae (two head arrow) in tunica media) (H&E. 400×). (**e**) G9: Section in aorta showing intact endothelium layer (two head arrow), smooth muscle (arrow) and elastic lamellae (arrowhead) in tunica media (M). (H&E. 400×).

## 4. Conclusions

In summary, the results demonstrated the synergic hypolipidemic and immunomodulatory activity of Lactobacillus and *Spirulina platensis* supplemented to mice fed a regular or high-cholesterol diet. Our findings showed that *Lactobacillus* spp., *Spirulina*, and their combination resulted in a significant decrease in body weight, serum cholesterol, LDL-c, and TG, and an increase in HDL-c in both mice fed on regular or high-cholesterol diets. Moreover, the results revealed that they also stimulated the innate immunity represented in both monocyte and neutrophil cells in mice fed regular diets, while reducing immunity in mice fed on high-cholesterol diets due to the reduction in the inflammation caused by hyperlipedimia. The beneficial effects on weight, lipid profiles and immunomodulation appear to occur when probiotics are consumed for 8 weeks. We can conclude that treatment with a combination of Lactobacillus and Spirulina has similar results to that of rosuvastatin. We propose that *Lactobacillus* has the capability to play an important role in the preparation of functional foods with health-promoting effects. Further studies are required to investigate the hypolypedimic and immunomodulatory effects of these bacteria.

**Author Contributions:** R.A.H. conceptualization, writing, figure draw, statistical analysis, and review; H.A.H. conceptualization, reviewing; M.L.S. upervision, methodology, reviewing; S.K. methodology; R.H.A. and I.A. resources; H.M. source of bacteria used; A.A. methodology, tables and figures, writing review and editing. All authors have read and agreed to the published version of the manuscript.

**Funding:** This research received no external funding.

**Institutional Review Board Statement:** Not applicable.

**Informed Consent Statement:** Not applicable.

**Conflicts of Interest:** The authors declare no conflict of interest.

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
