# Peer review of "Synergistic Hypolipidemic and Immunomodulatory Activity of Lactobacillus and Spirulina platensis"

_fermentation, doi:10.3390/fermentation8050220_

Round 1
Reviewer 1 Report
The manuscript has been improved compared to the previous version, but it still needs further corrections. Referring to the Authors' responses:
"The percentage of neutrophils and lymphocytes are totally different between humans and mice."
I am glad that you noticed this as you previously provided a link to human blood reference data.
Nevertheless, the total content of eosinophils and basophils does not exceed 10% in healthy humans or mice, whereas according to your data it exceeded 30% in control mice. Such results require some additional explanation. It is not known if such data were obtained as a result of incorrect calculations or for other reasons.
"Anyway, I have removed the percentage of individual leukocytes from the table and left the total leukocyte count to avoid any uncertainty."
Indeed, the percentages of individual leukocytes were removed from the table, but the description of them remained in some places in the text, for example in Material and Methods. Please remove it. In this situation, the "Results and discussion" section should also be redrafted and the discussion of the increase in neutrophils and monocytes should not be included in the subsection “Complete blood count”, but rather in “Analysis of myeloid cells with flow cytometry”.
Moreover, I suggest that the Authors carefully analyse and correct all the text, as it still contains errors, e.g. in some places rats are mentioned (L 158-159, 378, 397), while the experiment was carried out on mice.
Author Response
Comments and Suggestions for Authors
The manuscript has been improved compared to the previous version, but it still needs further corrections. Referring to the Authors' responses:
"The percentage of neutrophils and lymphocytes are totally different between humans and mice."
I am glad that you noticed this as you previously provided a link to human blood reference data.
Nevertheless, the total content of eosinophils and basophils does not exceed 10% in healthy humans or mice, whereas according to your data it exceeded 30% in control mice. Such results require some additional explanation. It is not known if such data were obtained as a result of incorrect calculations or for other reasons.
"Anyway, I have removed the percentage of individual leukocytes from the table and left the total leukocyte count to avoid any uncertainty."
Indeed, the percentages of individual leukocytes were removed from the table, but the description of them remained in some places in the text, for example in Material and Methods. Please remove it. In this situation, the "Results and discussion" section should also be redrafted and the discussion of the increase in neutrophils and monocytes should not be included in the subsection “Complete blood count”, but rather in “Analysis of myeloid cells with flow cytometry”.
Moreover, I suggest that the Authors carefully analyse and correct all the text, as it still contains errors, e.g. in some places rats are mentioned (L 158-159, 378, 397), while the experiment was carried out on mice.
Many thanks for valuable reviewer’s comments
I removed the description of individual leukocytes from materials and methods section
The discussion of the increase in neutrophils and monocytes has been moved to Analysis of myeloid cells with flow cytometry section
The text has been corrected and rats were replaced by mice in the revised manuscript
Reviewer 2 Report
I am satisfied with the revised manuscript. I am suggesting to accept the manuscript.
Author Response
Comments and Suggestions for Authors
I am satisfied with the revised manuscript. I am suggesting to accept the manuscript.
Many thanks for valuable reviewer’s comments

Round 2
Reviewer 1 Report
The manuscript was substantively improved as compared to its previous version, but it still contains some unclear statements and many typing and punctuation errors, as well as errors in style (repetitions) and in in-text reference style. Therefore, the entire text requires general editorial correction.
Detailed comments:
L 19-22: This statement suggests that you mean all groups fed on a high cholesterol diet, whereas you probably mean G5. Please be precise. Moreover, please include units into brackets with numerical data – here and elsewhere in the text.
L 35-36: “resulted in similar results” – Please correct the phrase, for example “gave similar results”.
L 44, 145, 147, 148: Please explain abbreviations the first time you use them.
L 87: “according to (Hoda et al., [16]” – Please correct the reference style here and in other places in the manuscript, e.g. in L 323-324.
L 117, 119: “diet with (1 mg/kg BW) mixture (1:1) ratio of cell lysates”; “diet with (1 mg/kg BW) of S. platensis” – Please put the parentheses in the correct places or remove them – here and elsewhere in the text.
L 184-191: The second paragraph is an almost exact repetition of the previous one, only the numerical data has been changed. Similarly in lines: 216-227, 258-267, 294-304, 243-254. Such repetitions do not sound good.
L 243-244: “The deconjugation of bile and binding to bile acid in the small intestine has been postulated as a possible explanation for lactic acid bacteria’s excretion of fecal bile acid” – This statement is unclear. I suggest: "The deconjugation of bile and binding to bile acid by lactic acid bacteria in the small intestine has been postulated as a possible explanation for the increased fecal excretion of bile acids”.
L 376-377: This statement contains an error, since there are some statistical differences in MCHC.
Author Response
Comments and Suggestions for Authors
The manuscript was substantively improved as compared to its previous version, but it still contains some unclear statements and many typing and punctuation errors, as well as errors in style (repetitions) and in in-text reference style. Therefore, the entire text requires general editorial correction.
Detailed comments:
L 19-22: This statement suggests that you mean all groups fed on a high cholesterol diet, whereas you probably mean G5. Please be precise. Moreover, please include units into brackets with numerical data – here and elsewhere in the text.
Thanks for valuable reviewer’s comments
High cholesterol diet was replaced by positive control group (G5)
Units were included into brackets with numerical data in whole manuscript
L 35-36: “resulted in similar results” – Please correct the phrase, for example “gave similar results”.
Thanks for valuable reviewer’s comments
It was corrected in revised manuscript
L 44, 145, 147, 148: Please explain abbreviations the first time you use them.
Thanks for valuable reviewer’s comments
Abbreviations were explained in revised manuscript
L 87: “according to (Hoda et al., [16]” – Please correct the reference style here and in other places in the manuscript, e.g. in L 323-324.
Thanks for valuable reviewer’s comments
References have been corrected in revised manuscript
L 117, 119: “diet with (1 mg/kg BW) mixture (1:1) ratio of cell lysates”; “diet with (1 mg/kg BW) of S. platensis” – Please put the parentheses in the correct places or remove them – here and elsewhere in the text.
Thanks for valuable reviewer’s comments
Parentheses were removed in revised manuscript

This manuscript is a resubmission of an earlier submission. The following is a list of the peer review reports and author responses from that submission.
Round 1
Reviewer 1 Report
The Authors of this study examined the effect of Lactobacillus casei, Lactobacillus plantarum, and Spirulina platensis as compared with rosuvastatin effect on selected blood parameters, as well as cardiac and splenic tissues in mice fed normal or high-cholesterol diet. Unfortunately, this manuscript does not meet the standards of scientific work because the Authors do not present the results of the statistical analysis in the “Results and discussion”. According to the “Statistical analysis” section the obtained results were subjected to statistical analysis using one-way analysis of variance (ANOVA). However, no results of this analysis were presented in tables or graphs, only mean values and standard errors were presented. Moreover, it is not clear which groups were compared to each other using this test: all groups together (G1-G9) or G1-G4 and G5-G9 separately? For which results statistically significant differences were assumed and at what p-value?
Additionally, the results of relative numbers of neutrophils, lymphocytes and monocytes presented in Tables 1 and 2 raise many doubts. Firstly, the percentage values of these leukocytes are inconsistent and do not correspond to reference data. For example, the sum of percentages of neutrophils, lymphocytes and monocytes in G1 is 60%, which is in fact impossible, because in this situation the sum of percentages of eosinophils and basophils will be as much as 40%. These results in other groups are also questionable, for example in G5 the sum of neutrophil, lymphocyte and monocyte percentages amounts to 105%. Secondly, the Authors have presented in the text an increase in eosinophil number as influenced by experimental treatments (L 316 and 337), but such results were not presented in tables. Moreover, in “Materials and Methods” only neutrophil percentage measurement is mentioned, not percentages of other leukocytes.
Detailed remarks:
Abstract: There is no reference to rosuvastatin-fed mice in the presentation of the results.
L17: It should be “mice fed on regular or high-cholesterol diet” instead of “mice fed on regular and high-cholesterol diet”.
L32: It should be “Spirulina platensis”.
Material and methods
L 141-142: The procedure of blood collection should be described in more detail, including the anaesthesia method and the information when the blood was collected, in what quantity and how it was prepared for testing.
L 142-144: The names of the used methods should be included in this description.
L 146: The euthanasia method should be provided.
L 154-156: Why were blood samples collected using two different anticoagulants? This might cause some variation in the results as anticoagulants were added to blood in different proportions.
L 163-169: The heading of the section should refer to the parameters that were tested rather than to the test method. Additionally, this paragraph content does not clearly explain the purpose of the analyses undertaken.
L 178-181 and Fig. 1: The abbreviation for "gram" is "g", not "gm".
Results and discussion: The Authors did not refer to the data obtained on the 15th day of the experiment in this section.
L 293-295: This reference may be misleading because Nielsen et al. [49] used a butyrate producer (Butyrivibrio fibrisolvens) as a probiotic, which changed the SCFA profile in the caecum. In the present experiment, the mechanism of decrease in LDL-C may not be exactly the same.
L 327: Here the Authors described the G9 group as the mice receiving commercial lovastatin, which is not true because in other places they defined this group as receiving rosuvastatin.
Conclusions
L 443-445: On what basis do the Authors stated that the preparations used had an immunomodulatory effect?
L 447: “They have fewer side effects than statins” statement is not relevant to these studies (which did not examine side effects of used preparations), so it should not be included in the “Conclusions”.
Author Response
Reviewer 1
The Authors of this study examined the effect of Lactobacillus casei, Lactobacillus plantarum, and Spirulina platensis as compared with rosuvastatin effect on selected blood parameters, as well as cardiac and splenic tissues in mice fed normal or high-cholesterol diet. Unfortunately, this manuscript does not meet the standards of scientific work because the Authors do not present the results of the statistical analysis in the “Results and discussion”. According to the “Statistical analysis” section the obtained results were subjected to statistical analysis using one-way analysis of variance (ANOVA). However, no results of this analysis were presented in tables or graphs, only mean values and standard errors were presented. Moreover, it is not clear which groups were compared to each other using this test: all groups together (G1-G9) or G1-G4 and G5-G9 separately? For which results statistically significant differences were assumed and at what p-value?
Thanks for reviewer’s comments
Significance of variance and p-value was added to all tables and figures at p < 0.05.
All groups (G1-G9) were compared together. All figures compared (G1-G9).Table 1 and Table 2 was joined in one table that compare all groups(G1-G9)
Additionally, the results of relative numbers of neutrophils, lymphocytes and monocytes presented in Tables 1 and 2 raise many doubts. Firstly, the percentage values of these leukocytes are inconsistent and do not correspond to reference data. For example, the sum of percentages of neutrophils, lymphocytes, and monocytes in G1 is 60%, which is in fact impossible, because in this situation the sum of percentages of eosinophils and basophils will be as much as 40%. These results in other groups are also questionable, for example in G5 the sum of neutrophil, lymphocyte, and monocyte percentages amounts to 105%. Secondly, the Authors have presented in the text an increase in eosinophil number as influenced by experimental treatments (L 316 and 337), but such results were not presented in tables. Moreover, in “Materials and Methods” only neutrophil percentage measurement is mentioned, not percentages of other leukocytes.
Thanks for reviewer’s comments
- The percentage of neutrophils, lymphocytes and monocytes has wide range:
(Neutrophils: 40% to 60%, Lymphocytes: 20% to 40%, Monocytes: 2% to 8%)
https://www.ucsfhealth.org/medical-tests/blood-differential-test
This lies in the same range of our results
- Eosinophil number was removed from text.
- Lymphocyte%, Monocyte % was added in the materials and methods of revised manuscript
Detailed remarks:
Abstract: There is no reference to rosuvastatin-fed mice in the presentation of the results.
Thanks for reviewer’s comments
Rosuvastatin-fed mice was referred to in abstract
L17: It should be “mice fed on regular or high-cholesterol diet” instead of “mice fed on regular and high-cholesterol diet”.
Thanks for reviewer’s comments
It was corrected in revised manuscript.
L32: It should be “Spirulina platensis”.
Thanks for reviewer’s comments
It was corrected in revised manuscript.
Material and methods
L 141-142: The procedure of blood collection should be described in more detail, including the anaesthesia method and the information when the blood was collected, in what quantity and how it was prepared for testing.
Thanks for reviewer’s comments
Method of anaethesia and blood collection was added to revised manuscript.
L 142-144: The names of the used methods should be included in this description.
Thanks for reviewer’s comments
The names of used methods were included in revised manuscript.
L 146: The euthanasia method should be provided.
Thanks for reviewer’s comments
The euthanasia method was provided
L 154-156: Why were blood samples collected using two different anticoagulants? This might cause some variation in the results as anticoagulants were added to blood in different proportions.
Thanks for reviewer’s comments
I have specified one method (EDTA).
L 163-169: The heading of the section should refer to the parameters that were tested rather than to the test method. Additionally, this paragraph content does not clearly explain the purpose of the analyses undertaken.
Thanks for reviewer’s comments
The heading was changed to Analysis of Myeloid Cells with Flow Cytometry
Reason of analysis of Myeloid Cells with Flow Cytometry was added to revised manuscript.
L 178-181 and Fig. 1: The abbreviation for "gram" is "g", not "gm"
Thanks for reviewer’s comments
Gram was changed to g
Results and discussion: The Authors did not refer to the data obtained on the 15th day of the experiment in this section.
Thanks for reviewer’s comments
Data obtained on the 15th day was added to the revised manuscript.
L 293-295: This reference may be misleading because Nielsen et al. [49] used a butyrate producer (Butyrivibrio fibrisolvens) as a probiotic, which changed the SCFA profile in the caecum. In the present experiment, the mechanism of decrease in LDL-C may not be exactly the same.
Thanks for reviewer’s comments
Thanks for your advice, I removed that reference
L 327: Here the Authors described the G9 group as the mice receiving commercial lovastatin, which is not true because in other places they defined this group as receiving rosuvastatin.
Thanks for reviewer’s comments
Commercial lovastatin was replaced by rosuvastatin
Conclusions
L 443-445: On what basis do the Authors stated that the preparations used had an immunomodulatory effect?
Thanks for reviewer’s comments
I justified the immunomodulatory effect by stating that the results revealed that both Lactobacillus spp., Spirulina, and its consortium stimulated the innate immunity represented in both monocyte and neutrophil cells in normal mice, while they reduced them in mice fed on high cholesterol diet as they reduce the inflammation caused by hyperlipedimia (Table 2).
L 447: “They have fewer side effects than statins” statement is not relevant to these studies (which did not examine side effects of used preparations), so it should not be included in the “Conclusions”.
Thanks for the reviewer’s comments
The statement was removed.
Reviewer 2 Report
However, the article is well written but I donot feel that it is a suitable article for fermentation journal. The scope of article is not matched.
My other comments are:
- In introduction section, authors need to mention about knowledge gap and how the present investigation will fulfill the knowledge gap ! Objective of investigation and novelty of investigation need to mention in a comprehensive way in introduction.
- In x axis of graphs data points are not required. Tukys post-hoc analysis is necessary to describe significant or not-significant result.
- All scientific names are needed to write in italic in whole manuscript.
- In figure 5, figure 4 and figure 3, meaning of error bar needs to be together with title of figure.
- Figures donot need border line.
- It is necessary to write the whole article according to format of journal.
Author Response
Reviewer 2
However, the article is well written but I don’t feel that it is a suitable article for fermentation journal. The scope of article is not matched.
My other comments are:
- In introduction section, authors need to mention about knowledge gap and how the present investigation will fulfill the knowledge gap! Objective of investigation and novelty of investigation need to mention in a comprehensive way in introduction.
Thanks for reviewer’s comments
Objective of investigation and novelty of investigation was demonstrated in the introduction.
- In x axis of graphs data points are not required. Tukys post-hoc analysis is necessary to describe significant or not-significant result.
Thanks for reviewer’s comments
Data points were removed. Significance of variance was added to all tables and figures at p < 0.05
- All scientific names are needed to write in italic in whole manuscript.
Thanks for reviewer’s comments
All scientific names were rewritten in italic
- In figure 5, figure 4 and figure 3, meaning of error bar needs to be together with title of figure.
Thanks for reviewer’s comments
Meaning of error bar was placed with title of figure.
- Figures donot need border line.
Thanks for reviewer’s comments
The border line was removed
- It is necessary to write the whole article according to format of journal.
Thanks for the reviewer’s comments
The article was rewritten according to format of journal
Round 2
Reviewer 1 Report
Some issues in the manuscript have been improved in accordance with the requirements, but an important question still requires correction or explanation:
Regardless of the percentage ranges of the individual leukocyte groups, the sum of the percentages of all leukocytes in an individual is always 100%. So, if you specify that the animals in G1 group on day 57 had 33% of neutrophils, 26.3% of lymphocytes and 1.66% of monocytes, it means that the remaining leukocytes (eosinophils and basophils) were 100%-60.96% = 39,04%, which is far too much compared to the reference data. These results are highly questionable, even if you did not analyse eosinophils and basophils separately. There are similar uncertainties also in other groups, so I cannot consent to the publication of such results.
Reviewer 2 Report
Response from authors are satisfactory. I suggest to accept the manuscript.